# Association of Metabolic and Endocrine Disorders with Bovine Ovarian Follicular Cysts

**DOI:** 10.3390/ani13213301

**Published:** 2023-10-24

**Authors:** Xiaoling Xu, Jiahua Bai, Kexiong Liu, Linli Xiao, Yusheng Qin, Meihong Gao, Yan Liu

**Affiliations:** Institute of Animal Husbandry and Veterinary Medicine, Beijing Academy of Agriculture and Forestry Sciences, Beijing 100097, China; xu_xiaoling1980@163.com (X.X.); bai_jiahua@126.com (J.B.); liukexiong2023@163.com (K.L.); xiao1990linli@163.com (L.X.); blackberrysheng@163.com (Y.Q.); gaomeihongwen@163.com (M.G.)

**Keywords:** metabolic disorder, RNA sequencing, ovarian steroidogenesis pathway, follicular cysts, bovine

## Abstract

**Simple Summary:**

Ovarian follicular cysts are one of the most common reproductive disturbances in dairy cows. However, the exact mechanism underlying this disorder is not clear enough and remains difficult to identify. Here, we found that there was a disrupted secretion of hormone and abnormal mRNA expression of corresponding receptors in bovine cystic follicles compared with control follicles. Further KEGG enrichment analyses of transcriptome indicated that the differential expressed genes were significantly enriched in the ovarian steroidogenesis pathway. Based on the findings of the study, metabolic and endocrine disorders were associated with bovine ovarian follicular cysts.

**Abstract:**

After estrus, when mature follicles fail to ovulate, they may further develop to form follicular cysts, affecting the normal function of ovaries, reducing the reproductive efficiency of dairy cows and causing economic losses to cattle farms. However, the key points of ovarian follicular cysts pathogenesis remain largely unclear. The purpose of the current research was to analyze the formation mechanism of ovarian follicular cysts from hormone and gene expression profiles. The concentrations of progesterone (P_4_), estradiol (E_2_), insulin, insulin-like growth factor 1 (IGF1), leptin, adrenocorticotropic hormone (ACTH) and ghrelin in follicle fluid from bovine follicular cysts and normal follicles were examined using enzyme-linked immunosorbent assay (ELISA) or 125I-labeled radioimmunoassay (RIA); the corresponding receptors’ expression of theca interna cells was tested via quantitative reverse transcription polymerase chain reaction (RT-qPCR), and the mRNA expression profiling was analyzed via RNA sequencing (RNA-seq). The results showed that the follicular cysts were characterized by significant lower E_2_, insulin, IGF1 and leptin levels but elevated ACTH and ghrelin levels compared with normal follicles (*p* < 0.05). The mRNA expressions of corresponding receptors, *PGR*, *ESR1*, *ESR2*, *IGF1R*, *LEPR*, *IGFBP6* and *GHSR*, were similarly altered significantly (*p* < 0.05). RNA-seq identified 2514 differential expressed genes between normal follicles and follicular cysts. The Kyoto Encyclopedia of Genes and Genomes (KEGG) analysis linked the ovarian steroidogenesis pathway, especially the *STAR*, *3β-HSD*, *CYP11A1* and *CYP17A1* genes, to the formation of follicular cysts (*p* < 0.01). These results indicated that hormone metabolic disorders and abnormal expression levels of hormone synthesis pathway genes are associated with the formation of bovine ovarian follicular cysts.

## 1. Introduction

Ovarian cysts are the most common reproductive dysfunction in high-producing dairy cows, generating significant economic loss to the dairy industry by reducing conception rate, lengthening nonproduction days and raising the replacement rate owing to infertility [1,2,3]. The definition of ovarian cysts in dairy cows has always been controversial in different research projects. Previously, cattle ovarian cysts were defined as follicular-like structures, generally with a diameter of at least 25 mm present on one or both ovaries and a duration of more than 10 days [4]. Most recently, ovarian follicular cysts or luteal cysts were defined as anovulatory structures on the ovary, with a follicle cavity more than 20 mm in diameter and with an absence of corpus luteum [5]. Ovarian cysts can be classified functionally as follicular or luteal, and their difference is the thickness of the follicular wall. Follicular cysts have a relatively thin wall (≤3 mm), while luteal cysts have thicker walls (>3 mm) [5,6,7].

The prevalence of ovarian cysts in dairy cows may range from 2.7% to 30%, and a greater proportion of follicular cysts has been observed [8,9]. The exact mechanism regarding follicular cysts in dairy cows has not yet been completely explicit. Studies over many years have shown that the formation of ovarian follicular cysts is mainly the result of endocrine disturbance within the hypothalamic–pituitary–gonadal axis (HPGA) caused by endogenous and/or exogenous factors [4,10,11]. The most widely accepted hypothesis is that luteinizing hormone (LH) release from the hypothalamus–pituitary is changed and the preovulatory LH surge is either missing, is deficient in magnitude or happens at the incorrect time during the maturation of the dominant follicle [11,12,13,14]. Meanwhile, progesterone (P_4_) is involved in the formation of ovarian cysts, and there is a strong association between intermediate concentrations of P_4_ in peripheral blood and the occurrence of ovarian follicular cysts [4]. Most cysts are accompanied by a decrease in P_4_, which promotes the development of the cyst [15]. Molecular analysis of bovine follicular cyst disease pathogenesis has revealed that ovarian cysts exhibit partial disrupted steroid receptor patterns related to follicle-stimulating hormone receptor (*FSHR*), progesterone receptor (*PGR*), LH/choriogonadotropin receptor (*LHCGR*) and estrogen receptor (*ESR*) [1,16,17].

Reproductive success is a key component of lifetime efficiency, and negative energy balance (NEB) in cattle is considered an important contributor to reproductive disorders in dairy cows [18]. NEB can also lead to the formation of ovarian follicular cysts [11]. Imbalance between energy acquisition through feed intake and energy consumption through high milk production during early postpartum lactation results in NEB, which is often accompanied by hormonal and metabolic alterations that affect ovarian normal function [19,20]. During NEB, the concentrations of insulin-like growth factor 1 (IGF1), insulin [19] and leptin [21,22] in serum are decreased. Zulu et al. [23] found that low systemic concentrations of IGF-1 in early postpartum may lead to ovulation failure and the subsequent development of cystic follicles, while Vanholder et al. [24] discovered that reduced serum insulin content in early postpartum may play a part in ovarian function disturbance (i.e., cyst formation). Spicer [25] thought that above a certain threshold level, leptin can act as a trigger for initiating hypothalamo–pituitary gonadotropin secretion. In a moderate to high level of leptin circumstances, just as in obesity, leptin regulates ovarian steroidogenesis [25]. Previous studies indicate that the molecular mechanism of the formation of ovarian follicular cysts in dairy cows is complicated. However, hormone metabolic profiles and RNA-seq have not yet been widely employed for the determination of large-scale gene expression patterns to explain the molecular mechanism of ovarian follicular cyst formation. 

Therefore, the present study aimed to investigate the formation mechanism of ovarian follicular cysts from hormonal and gene expression patterns via deep sequencing of the transcriptome to identify differential expressed genes and their associated biological pathways that were important in ovarian follicular cysts in cattle. These findings set the foundation for further insight into the formation mechanisms, the prevention and treatment of cattle ovarian follicular cysts.

## 2. Materials and Methods

### 2.1. Collection of Ovaries

The ovaries with follicular cysts (*n* = 30) and control follicles (*n* = 19) of cows were collected from a Beijing local abattoir and transported to our laboratory within 2 h in the 37 °C prewarmed phosphate-buffered saline (PBS) with Penicillin-Streptomycin buffer (Biological Industries, Beit HaEmek, Israel). In the absence of active luteal tissue, a fluid-filled follicle structure, diameter >25 mm, with smooth thin and translucent walls (≤3 mm) was defined as a follicular cyst. Normal follicles (15 mm in diameter) with no gross morphological abnormalities were assigned to the control group.

### 2.2. Collection of Follicular Fluid and Theca Interna

Individual follicles were carefully dissected from the ovarian stroma with scissors and forceps after three-times PBS washing in 100 mm dishes. And the collection of follicular fluid and theca interna was according to the method published previously [26]. The collected follicular fluid was then centrifuged at 2000× *g* for 10 min and stored at −80 °C for further hormone detection; the theca interna was then frozen in liquid nitrogen and stored at −80 °C for subsequent RNA extraction and mRNA expression analysis.

### 2.3. Measurement of Hormone Concentration

The concentrations of estradiol (E_2_, B05PZB), progesterone (P_4_, P08PZB), insulin (F01PZB) and adrenocorticotropic hormone (ACTH, D14B) were measured using 125I-labeled radioimmunoassay kits (Beijing North Biotechnology Institute, Beijing, China) according to the operating instructions of the manufacturer. For the assays, the sensitivities of E_2_, P_4_, insulin and ACTH were 2 pg/mL, 2 ng/mL, 2 μIU/mL and 3 pg/mL, respectively, and the intra- and interassay coefficients of variation were 10.0% and 15.0%. ELISA was used to measure the concentrations of insulin-like growth factor 1 (IGF-1, DY291, R&D, Minneapolis, MN, USA), leptin (DLP00, R&D) and ghrelin (Jiancheng Bioengineering Institute, Nanjing, China) levels in follicular fluids, and assays were performed according to the operating instructions of the manufacturer. Briefly, 50 µL of standards or samples were added to the appropriate well of the microtiter plate precoated with antibody, gently mixed and incubated for 60 min at 37 °C. After washing, biotinylated anti-IgG and streptavidin–horseradish peroxidase (HRP) were added along with chromogen solutions A and B. Finally, the optical density (OD) at 450 nm was recorded using a Multiskan MK3 automatic microplate spectrophotometer (Thermo Fisher Scientific, Waltham, MA, USA). Hormone concentrations were calculated according to standard curves, and each experiment was performed, repeated independently at least three times.

### 2.4. Quantitative Reverse Transcription Polymerase Chain Reaction (RT-qPCR)

RNAzol reagent (RNAzol RT reagent, rn190; Molecular Research Center, Cincinnati, OH, USA) was applied to isolate the total RNAs. A NanoDrop 2000c spectrophotometer (Thermo Fisher Scientific) was used for qualitative analysis. Total RNA (0.5 µg) was used to perform first-strand cDNA synthesis via an Evo M-MLV RT kit (Accurate Biotechnology (Hunan) Co. Ltd., Changsha, China) according to the manufacturer’s instructions. All primers were designed on the basis of their gene sequences (Table 1). cDNA was then quantified with RT-qPCR using a CFX Connect Real-Time PCR System (Bio-Rad, Hercules, CA, USA). The qPCR was carried out in a 20 µL reaction volume with an iTaq™ universal SYBR^®^ Green super mix reagent kit (Bio-Rad Laboratories, Inc., Hercules, CA, USA) as follows: 95 °C for 5 min, followed by 40 cycles of 95 °C for 5 s, 60 °C for 15 s and 72 °C for 15 s. Bovine *GAPDH* was selected as an internal control. Three technical replicates were performed on each cDNA, and average Ct value was used for further analysis. Amplification efficiencies were close to 100%. Relative expression values were calculated using the 2^−ΔΔCt^ method.

### 2.5. RNA Extraction, Library Preparation, Sequencing and Bioinformatics Analysis

Total RNA was extracted from the theca interna of spontaneous follicular cysts (*n* = 5) and control follicles (*n* = 4) of cows using TRIzol reagent (Invitrogen, Carlsbad, CA, USA) according to the manufacturer’s instructions, and RNA was purified using an miRNeasy kit (Qiagen, Hilden, Germany). Sequencing and bioinformatics analysis were conducted by Beijing Genomics Institute (BGI; Beijing, China). Detailed procedures had been published previously [27,28]. Raw reads had been deposited in the National Center for Biotechnology Information (NCBI) Sequence Read Archive database (https://www.ncbi.nlm.nih.gov/sra, accessed on 19 January 2020) under accession number PRJNA602176. Raw RNA-seq data were filtered into clean reads, followed by mapping against the Bos taurus reference genome (mm10) using HISAT. Gene expression levels were quantified using the RSEM software package [29]. Differential expressed genes (DEGs) between the control and follicular cyst group were identified using fold change ≥2 and false discovery rate (FDR) ≤ 0.001 as criteria. Gene Ontology (GO) annotation was used to map all DEGs to GO terms in the database (http://www.geneontology.org/, accessed on 24 March 2020), and GO terms with Q-values (corrected *p*-value) ≤ 0.05 defined DEGs as significantly enriched. The Kyoto Encyclopedia of Genes and Genomes (KEGG) database was used to perform pathway enrichment analysis of DEGs, and pathway terms with Q-values ≤ 0.05 were defined as significantly enriched [30].

### 2.6. Statistical Analysis

Two-tailed Student’s *t* tests with IBM SPSS Statistics for Windows version 20.0 were used to analyze the data from the control and follicular cyst groups (IBM Corp., Armonk, NY, USA), and the results were presented as mean percentages ± standard error of the mean (SEM). Statistical significance was defined at *p* < 0.05.

## 3. Results

### 3.1. Hormonal and Metabolic Profile of Follicular Fluid

Concentrations of hormones in the follicular fluids of follicular cysts (*n* = 30) and control follicles (*n* = 19) were assayed, and the hormonal profiles are shown in Figure 1. Firstly, follicular cyst follicles had extremely significant lower E_2_, insulin, IGF1 and leptin levels compared with normal follicles (*p* < 0.01). Secondly, follicular cyst follicles had extremely significant higher ghrelin and ACTH levels compared with normal follicles (*p* < 0.01). There was no significant difference in P_4_ between follicular cyst follicles and normal follicles.

### 3.2. Relative mRNA Levels of Corresponding Hormone Receptors

The relative mRNA expression levels of corresponding hormone receptors were tested via qRT-PCR, and the results are shown in Figure 2. The transcription levels of *ESR1*, *IGFBP6* and *GHSR* were significantly higher in cystic follicles than in control follicles (*p* < 0.05). The transcription levels of *ESR2* and *PGR* were significantly lower in cystic follicles than in control follicles (*p* < 0.05). The transcription levels of *IGF1R* and *LEPR* were extremely significantly lower in cystic follicles than in control follicles (*p* < 0.01). There was no significant difference in INSR between cystic follicles and control follicles (*p* > 0.05).

### 3.3. Transcriptome Analysis of Ovarian Follicular Cysts

In this study, nine RNA samples from the theca interna of follicular cysts and controls were subjected to deep sequencing using an Illumina HiSeq1500 platform (Illumina, San Diego, CA, USA). A total of 2514 genes displayed differential expression between the follicular cyst and control groups, among which 2077 and 437 DEGs were up- and down-regulated, respectively (Figure 3). 

### 3.4. Clustering Analysis of DEGs

In order to identify the functions of DEGs in the follicular cyst vs. control groups, GO analysis was performed based on three functional categories, biological process (BP), cellular component (CC) and molecular function (MF), and the results are shown in Figure 4. The GO enrichment analysis revealed that many DEGs were implicated in the regulation of the response to stress, cell proliferation, ovulation cycle process, hormone metabolic process and regulation of the reproductive process. 

### 3.5. Pathway Enrichment Analysis of DEGs

Genes usually interact with each other to play roles in specific biological functions. Thus, the pathway enrichment analysis of DEGs was performed based on the KEGG database, and the DEGs detected in the comparisons of the follicular cyst group vs. control were mapped to KEGG metabolic and regulatory pathways with a correct *p*-value cutoff of *p* < 0.05. The related KEGG enrichment results are shown in Figure 5. From the pathway analysis, the ovarian steroidogenesis pathway, cAMP signaling pathway, cytokine–cytokine receptor interaction, cell cycle and PI3K-Akt signaling pathway took part in the formation of follicular cysts.

### 3.6. Functions of DEGs via Pathways Analysis

Pathway analysis showed that the ovarian steroidogenesis pathway was associated with the formation of follicular cysts. Thus, the expression of key genes involved in this signaling pathway was further analyzed (Table 2). The results showed that steroidogenic acute regulatory protein (*STAR*), hydroxy-delta-5-steroid dehydrogenase, 3-beta- and steroid delta-isomerase 1 (*3β-HSD*), cytochrome P450, family 11, subfamily A, polypeptide 1 (*CYP11A1*), cytochrome P450, family 17, subfamily A and polypeptide 1 (*CYP17A*1) were up-regulated significantly (*p* < 0.05). And the results were also further verified using RT-qPCR, and the same trends were acquired, as seen in Figure 6.

## 4. Discussion

In the current study, we performed hormonal and transcriptome profiles to analyze the mechanism of cattle ovarian follicular cysts formation. The high prevalence of ovarian cysts in high-yield dairy cows has negative effects on the reproductive performance and economic income of dairy farms. 

It is well known that dominant follicle selection is followed by ovulation [31]. Steroid hormones play a crucial role in ovarian development, differentiation and folliculogenesis. Antral follicle growth is induced by follicle-stimulating hormone (FSH) and associated with elevated E_2_ production; high E_2_ levels enhance hypothalamic gonadotropin-releasing hormone (GnRH) pulses, triggering the coming of LH surge [32]. However, reduced E_2_ levels and an absent preovulatory LH surge at the appropriate time during the maturation of the dominant follicle results in the formation of ovarian follicular cysts in cattle [1,33]. In our study, the hormonal profiles of follicular fluids showed that compared with control follicles, follicular cyst follicles had lower E_2_ and almost equal P_4_. The ratios of E_2_ to P_4_ (E/P) in the follicular fluids of ovarian cysts decreased significantly. Braw-Tal et al. [34] reported that preovulatory follicles were characterized by high E_2_ and low P_4_ concentrations, and the E/P ratio in these follicles was 42, but this ratio dropped sharply to a low level in follicular cysts. Follicular cysts could contain significantly lower P_4_ [17] or greater P_4_ [35] but significantly lower E_2_ levels [17,35], or they could tend to decrease in follicular fluids compared to preovulatory follicles [36]. An imbalance between E_2_ and P_4_ in intrafollicular fluids leads to the formation of ovarian cysts. 

The function of hypothalamic–pituitary and the growth and development of ovarian follicles may be influenced by NEB through metabolic and/or hormonal alteration. Insulin and IGF-1 are both considered as key mediators between bovine nutritional status and ovarian physiology function [37,38,39]. NEB and reduced IGF1 are associated with reduced fertility. In vitro and in vivo studies on cows show that insulin and IGF-1 stimulate synthesis and secretion of both E_2_ in granulosa cells and androgen in theca cells [40,41]. Our data indicated that the average insulin and IGF-1 concentrations in the follicular fluids of follicular cyst follicles were significantly lower than in control follicles. Therefore, reduced insulin and IGF-1 levels may have an influence on the follicular responsiveness to LH stimulation and then result in ovulation failure and further cyst formation. Ghrelin, the endogenous ligand of the growth hormone secretagogue receptor (GHS-R1a), plays an important role in the hypothalamo–pituitary–gonadal axis, and it inhibits the secretion of gonadotropin-releasing hormone (GnRH) in the hypothalamuses of rats [42,43]. Some studies evaluate the relationship between metabolic status and follicular cysts but seldom include ghrelin. This aspect would be interesting because ghrelin is related to NEB [44]. We found that the concentration of ghrelin was significantly higher in the follicular fluids of follicular cysts than in control follicles. Metabolic hormones such as positive signals (IGF1, leptin) and negative signals (ghrelin) serve as signaling molecules that regulate the activity of GnRH neurons in the hypothalamus and the function of the reproductive endocrine system [45]. ACTH could be referred to as the regulatory mechanisms associated with ovarian function, such as ovulation, ovarian steroidogenesis and luteal function [46]. Herein, ACTH was significantly higher in the follicular fluids of follicular cysts than in control follicles. It is reported that exogenous ACTH treatment can promote endogenous cortisol synthesis and secretion, thus leading to follicular cysts in cattle [47]. So, the disruption of metabolic homeostasis leads to the failure of ovulation and the formation of follicular cysts. 

Steroid hormones act through specific receptors which directly control the expression levels of specific gene complexes regulating the growth, development and differentiation of reproductive tissues and cells, as well as other metabolic processes [48]. Steroidogenesis, which is essential to maintaining normal ovarian physiology functions, involves complicated enzymatic pathways regulated by insulin and other key proteins [49]. Abnormal secretion of steroid hormones and metabolic factors can explain the formation of ovarian cysts, but the specific molecular mechanism remains unknown. Our data also showed that the mRNA expressions of *ESR*1, *ESR*2, *PGR*, *IGF1Rs* and *GHSR* were changed in follicular cysts. It has been previously demonstrated that the tiny imbalance in the expression of the two subtypes of *ESR* could be involved in the pathogenesis of follicular cysts in cattle [50]. Some researchers reported that follicular cysts appear to be associated with changes in the transcription of *IRs*, *IGFRs* [24], *PAPP-A* [51] and *HSD3B1* and LH receptor genes [52], as well as a decreased expression level of estrogen receptor β protein and a promoted expression level of estrogen receptor α protein [53]. The changes in the expression of *ER-β* may destroy the partial intraovarian paracrine/autocrine system, resulting in changed follicular development and steroidogenesis and finally the formation of follicular cysts [54]. The steroid receptor is associated with follicular health and the stage of development; any modifications in intrafollicular activity of steroid hormones decides the fate of a specific follicle, resulting in the formation of follicular cysts.

However, high-throughput studies on the formation of ovarian cysts are lacking. A previous microarray analysis investigated gene expression in granulosa cells from dominant and cystic follicles, revealing 163 DEGs, of which 19 were up-regulated and 144 were down-regulated [35]. As the development of advanced molecular genetics technologies, especially next-generation sequencing and bioinformatics, transcriptome sequencing (RNA-seq) provides a convenient platform for analyzing large-scale gene expression patterns in organisms [55]. Some studies have analyzed the transcriptome profiles of liver samples from lactating dairy cows divergent in NEB [56] and the anterior pituitary of heifers before and after ovulation [57]. In our study, large-scale gene expression patterns were used to analyze the molecular mechanism of ovarian follicular cysts. The results showed that the expressions of *STAR*, *3β-HSD*, *CYP11A1* and *CYP17A1* genes on the ovarian steroidogenesis pathway were significantly different in ovarian follicular cysts compared with the control follicles. The synthesis of steroid hormones is under strict control; subtle adapted secretion of steroids usually results in serious disorder. The cholesterol transport protein, *StAR*, promotes the translocation of cholesterol from the outer mitochondrial membrane to the inner mitochondrial membrane and the two steroidogenic enzymes, *3β-HSD* is critical for the conversion of cholesterol to pregnenolone [58,59,60], and *CYP11A1* controls the first and rate-limiting step of steroid biosynthesis through the cAMP-signaling pathway [61]. The gene expression pattern corresponded well to the hormone profiles. The results revealed that the ovarian steroidogenesis pathway was linked to the formation of ovarian cysts by changing the hormone profile.

## 5. Conclusions

The results presented herein showed that decreased E_2_, insulin, IGF1 and leptin levels and increased ghrelin and ACTH levels in the follicular fluids of follicles and changed expressions of corresponding receptors in the theca interna set a key pole in the process of ovarian follicular cysts. Further analysis of the transcriptome via RNA-seq found the up-expression of *STAR*, *3β-HSD*, *CYP11A1* and *CYP17A1* on the ovarian steroidogenesis pathway associated with cattle ovarian follicular cysts. Our current work comparing cystic and normal follicles greatly expands our knowledge in this area. However, coupled with previous studies, the main cause of cyst formation has not been completely understood. There are still some limitations to researching the clinical cases of spontaneous ovarian follicular cysts cows; the conditions affecting ovarian function may be different between the time of diagnosis and cysts formation. Therefore, further investigation is needed to determine cause-and-effect relationships. 

## Figures and Tables

**Figure 1 animals-13-03301-f001:**
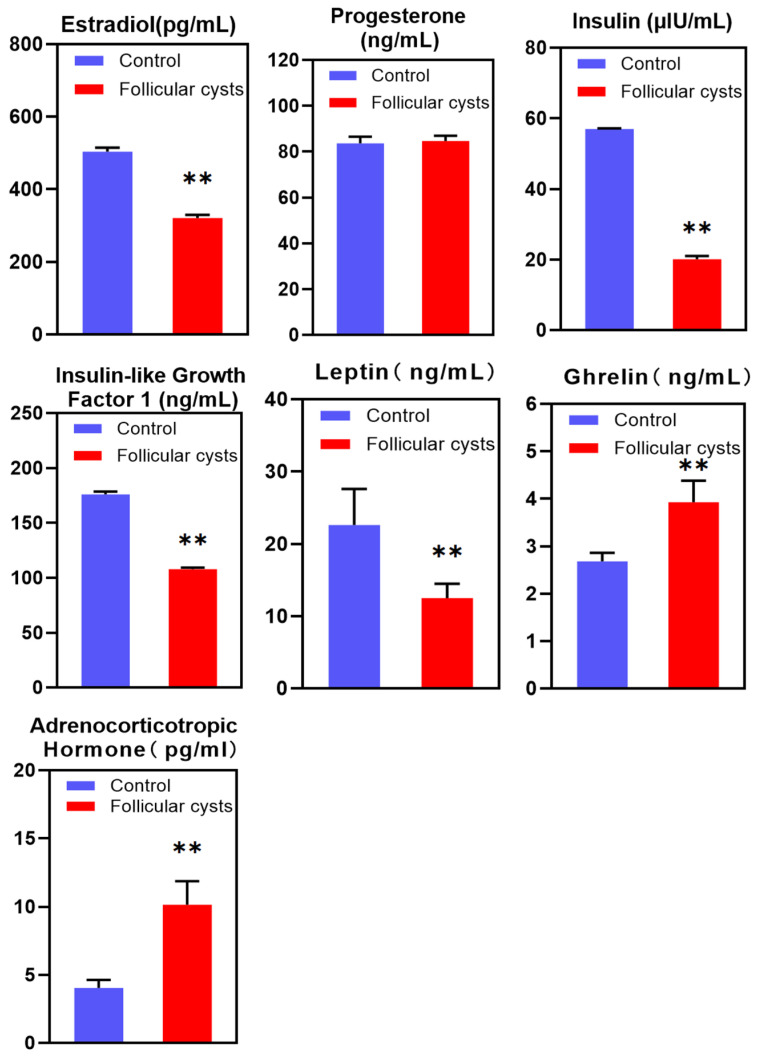
Concentrations of estradiol (E_2_), progesterone (P_4_), insulin, insulin-like growth factor 1 (IGF1), leptin, ghrelin and adrenocorticotropic hormone (ACTH) in the follicular fluid of follicular cyst and control follicles. ** Indicates statistically extremely significant (*p* < 0.01).

**Figure 2 animals-13-03301-f002:**
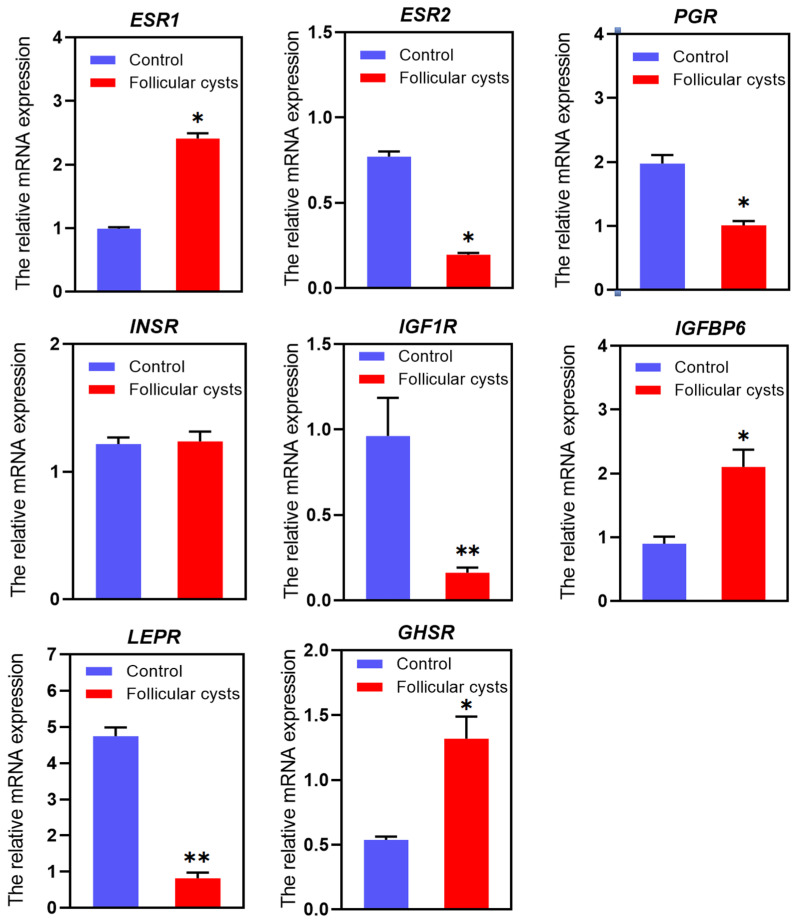
The relative mRNA expression of *ESR1*, *ESR2*, *PGR*, *INSR*, *IGF*1*R*, *IGFBP6*, *LEPR* and *GHSR* genes in follicular cyst and control follicles. * Indicates statistically significant (*p* < 0.05) and ** indicates statistically extremely significant (*p* < 0.01).

**Figure 3 animals-13-03301-f003:**
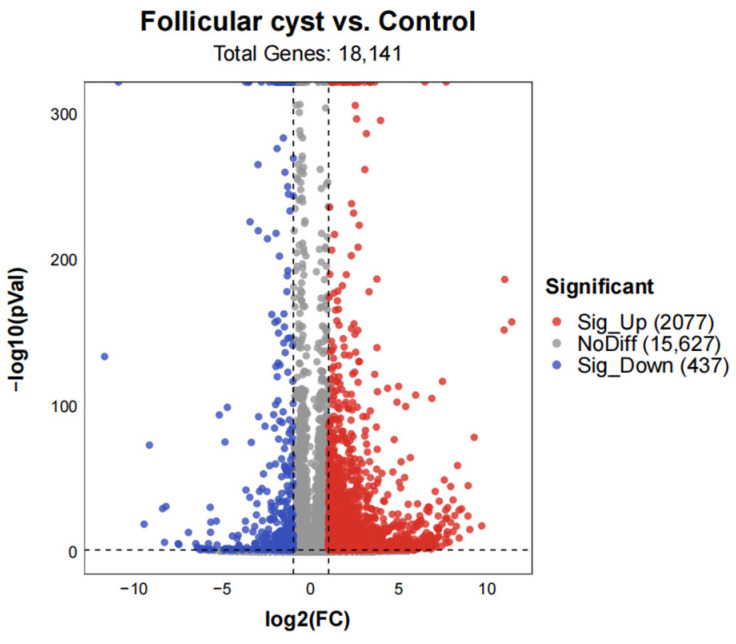
Volcano plot of DEGs identified in follicular cyst vs. control. *X* axis represents value A (log2-transformed mean expression level). *Y* axis represents value M (−log10-transformed fold change). Red dots represent up-regulated DEGs. Blue dots represent down-regulated DEGs. Gray points represent non-DEGs.

**Figure 4 animals-13-03301-f004:**
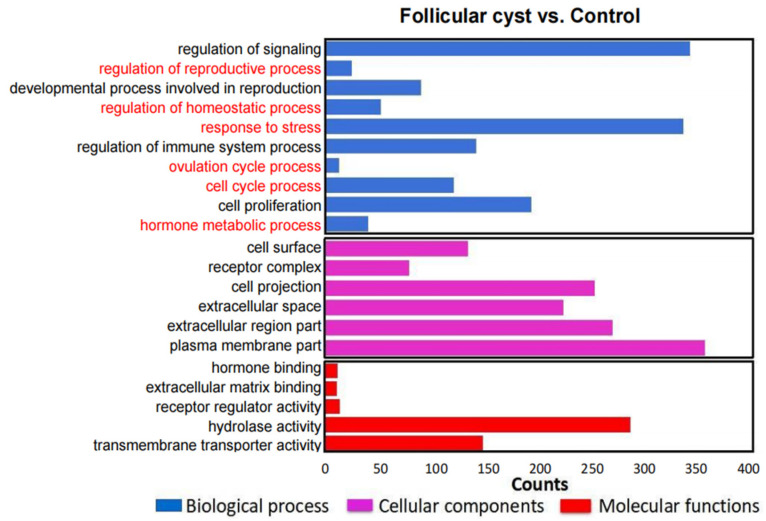
Gene Ontology (GO) enrichment analysis of DEGs for biological processes (BPs), cellular components (CCs) and molecular functions (MFs).

**Figure 5 animals-13-03301-f005:**
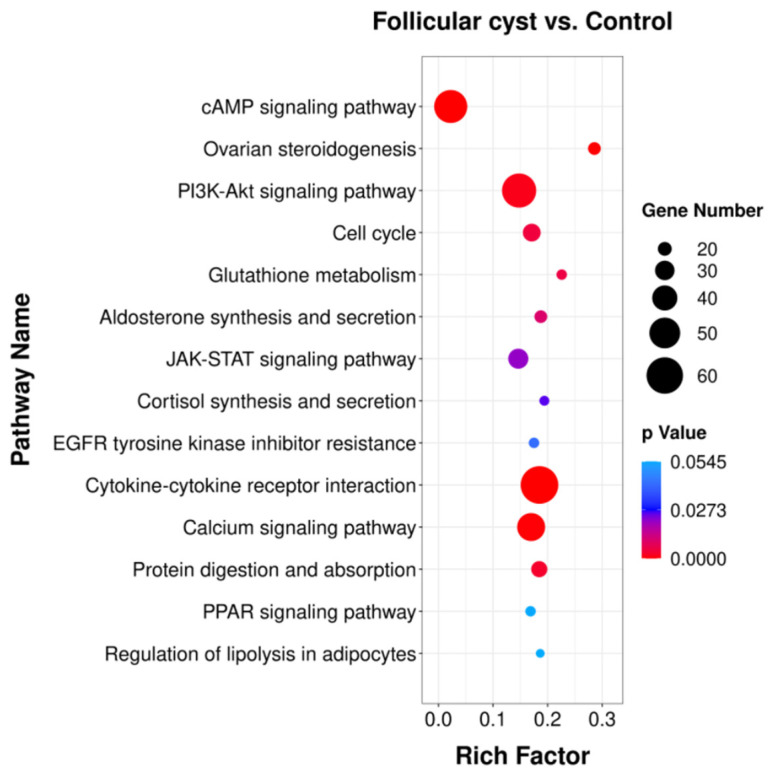
Statistics of pathway enrichment of DEGs in the follicular cyst group vs. control group. Rich factor is the ratio of differential expressed gene numbers annotated in this pathway term to all gene numbers annotated in this pathway term. Size of dots: number of genes; color of dots: range of *p*-values.

**Figure 6 animals-13-03301-f006:**
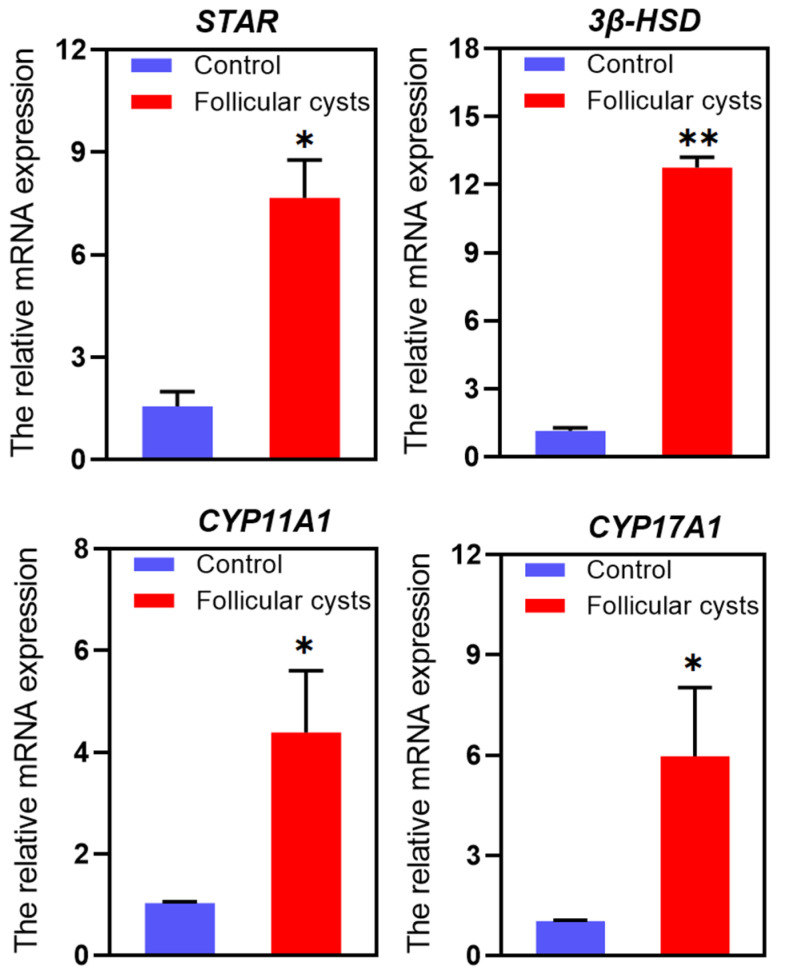
The relative mRNA expression of *STAR*, *3β-HSD*, *CYP11A1* and *CYP17A1* genes in follicular cyst and control follicles. * Indicates statistically significant (*p* < 0.05) and ** indicates statistically extremely significant (*p* < 0.01).

**Table 1 animals-13-03301-t001:** Primers for PCR amplification.

Genes	Primer Sequence	Accession Numbers	Product Length (bp)
*ESR1*	F: AGGGAAGCTCCTATTTGCTCC R: CGGTGGATGTGGTCCTTCTCT	AY538775	234
*ESR2*	F: AACCTGCTGATGCTCCTGTC R: CAAAGACTTGTTGCCGCGAA	NM_174051.3	150
*PGR*	F: GAGATCTTATAAGCATGTCAGTGG R: TCATGCAAGTTATCAAGAAGTTTT	NM_001205356.1	360
*INSR*	F: CGTGACAGACTATTACGTGCC R: CCCAATTCTCGCAGGAGTGT	XM_005208815.3	258
*IGF1R*	F: CCTCATCAGCTTCACCGTCTACT R: GCGTCCTGCCCGTCATACT	XM_606794.3	72
*IGFBP* *6*	F: ACACTGAGATGGGTCCCTGC R: AGAAGCCCCTTTGGTCACAA	NM_001040495.2	117
*LEPR*	F: CTGCTCCCCCAGAAAHACAG R: GCTGAGCTGACATTGGAGGT	XM_010803431	172
*GHSR*	F: CTCGTCATCCTGGTCATCTGGG R: AACTCGGTCGCTCGGCACTC	NM_001143736.2	125
*STAR*	F: GGAGGAGATGGCTGGAAGAAGGT R: TGCTGTAGCACTGGAATGGAAACA	NM_174189	174
*3β-HSD*	F: ACCTGGGAGTGACAATGATGGGAA R: TCTGGTGGCGGAAGGCAGATAGTA	NM_174343	161
*CYP1* *1A1*	F: CTACCAGGACCTGAGACGGA R: CCTGCCAGCATCTCCGTAAT	NM_176644.2	123
*CYP17A1*	F: TCCTGGCTGTCTTTCTGCTCA R: GTGTCCAATCATCACAGTCGT	BOVCYP17A	224
*GAPDH*	R: GTCATTGATGGCGACGATGT R: CTAACCTCGGGGAGAGCTTG	NM_001034034.2	100

**Table 2 animals-13-03301-t002:** Expression of key DEGs associated with the formation of ovarian follicular cysts.

Gene	Gene ID	Control (FPKM)	Follicular Cysts (FPKM)
*STAR*	281507	3.9	20.1
*3β-HSD*	281824	16.74	90.54
*CYP11A1*	338048	11.03	66.07
*CYP17A1*	281739	0.92	80.42

Note: FPKM means fragments per kilobase per million.

## Data Availability

None of the data were deposited in an official repository. All data generated during the study are available from the corresponding author by request.

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
