# Peer review of "Association of Metabolic and Endocrine Disorders with Bovine Ovarian Follicular Cysts"

_animals, 2023, doi:10.3390/ani13213301_

Round 1

Reviewer 1 Report

The article has clearly indicated significant questions to answer the ongoing challenges related to endocrine disorders. The study was comprehensively designed, and methods were appropriately described. I also did not find issues with the results, as they were beautifully described.

Author Response

Dear reviewer, thank you so much for taking your valuable time to review my manuscript.  Best wishes.

Reviewer 2 Report

The paper under review aimed to investigate the hormonal and gene expression patterns in ovarian follicular cysts in cattle. The topic is within the scope of the journal. The general study design is satisfactory. The evaluated paper contributes new information to the field. The study showed lower levels of E2, insulin, IGF1 and leptin. The mRNA expression of receptors , PGR, ESR1, ESR2, IGF1R, LEPR, IGFBP6, GHSR were altered significantly. A total 195 genes displayed differential expression between follicular cyst and control.

Comments

My main doubt concerns the interpretation of the results. It is not clear to what extent the hormonal and metabolic changes found in ovarian cysts are involved in their formation as the Authors suggests, or whether they are rather the result of them. The most widely accepted hypothesis explaining the formation of a cyst is that LH release from the hypothalamus-pituitary is insufficient. This should be discussed in more depth.

The study has limitations. Ovaries were collected from slaughter animals. Thus, it is not known at what time from calving the cows were. Ovarian cysts occur most often during the early postpartum period, when there is metabolic and hormonal imbalance. Metabolic and hormonal determinations in peripheral blood were not performed and it is not known to what extent the changes in the cystic and follicular fluid reflect the systemic situation. This should be highlighted as a limitations of the study as a separate paragraph in the discussion section.

The conclusions are too general and should be re-written. They should refer to the results obtained.

Author Response

Dear reviewer, thank you so much for taking your valuable time to review my manuscript.

Comments

  1. My main doubt concerns the interpretation of the results. It is not clear to what extent the hormonal and metabolic changes found in ovarian cysts are involved in their formation as the Authors suggests, or whether they are rather the result of them. The most widely accepted hypothesis explaining the formation of a cyst is that LH release from the hypothalamus-pituitary is insufficient. This should be discussed in more depth.

Dear reviewer, thanks for your professional comments. Frankly, the hormonal and metabolic changes found in ovarian follicular cysts indeed are the result of cysts for some extent. The disorder of the hypothalamic-pituitary- ovarian axis is a major dysfunction at the follicular level, leading to the formation of ovarian cysts. So, the secretion of steroid hormones and the expression of receptor genes in the ovaries can also affect the secretion of hypothalamic-pituitary hormones. The pre-ovulatory LH-surge is critical in the process of ovulation. The alteration of LH release from the pituitary can lead to the formation of ovarian cyst.    

  1. The study has limitations. Ovaries were collected from slaughter animals. Thus, it is not known at what time from calving the cows were. Ovarian cysts occur most often during the early postpartum period, when there is metabolic and hormonal imbalance. Metabolic and hormonal determinations in peripheral blood were not performed and it is not known to what extent the changes in the cystic and follicular fluid reflect the systemic situation. This should be highlighted as a limitations of the study as a separate paragraph in the discussion section.

Dear reviewer, thanks for your professional comments. This is indeed a common issue in current research in this field. Our previous study about this aspect showed that ovarian cysts occur most often during the early postpartum period, and the hormone concentrations changed in peripheral blood (previous published in Chinese). We highlighted the limitations in the Conclusion part, please see line 352-357. We thought the limitation could provide new ideas for subsequent studies.

  1. The conclusions are too general and should be re-written. They should refer to the results obtained.

Dear reviewer, thank you for your good suggestion. We have revised the conclusions, please see line 345-357 in the revised manuscript.

Reviewer 3 Report

The authors sought to analyze the formation mechanism of ovarian follicular cyst from hormone and gene expression profiles. To this aim, taking advantage of RIA, ELISA and molecular analysis, they evaluated the expression levels and abundance of several hormonal markers, including progesterone, insulin, leptin, adrenocorticotropic hormone and ghrelin, in follicle fluid from bovine follicular cysts and normal follicles were examined. The very interesting novelty of the paper relies on the high prevalence of ovarian cyst in high-yield dairy cows, which definitely has negative effects on the reproductive performance and economic benefits of dairy farms. The paper is well written, methodologically sounds and therefore suitable for the publication.

Author Response

Dear reviewer, thank you so much for taking your valuable time to review my manuscript. Best wishes for you.